# Epidemiology of Deltacoronaviruses (δ-CoV) and Gammacoronaviruses (γ-CoV) in Wild Birds in the United States

**DOI:** 10.3390/v11100897

**Published:** 2019-09-26

**Authors:** Francine C. Paim, Andrew S. Bowman, Lauren Miller, Brandi J. Feehan, Douglas Marthaler, Linda J. Saif, Anastasia N. Vlasova

**Affiliations:** 1Food Animal Health Research Program, The Ohio Agricultural Research and Development Center, Department of Veterinary Preventive Medicine, The Ohio State University, 1680 Madison Ave., Wooster, OH 44691, USA; franchimelo@gmail.com; 2Department of Veterinary Preventive Medicine, College of Veterinary Medicine, The Ohio State University, 1920 Coffey Rd, Columbus, OH 43210, USA; bowman.214@osu.edu; 3College of Public Health, The Ohio State University, 250 Cunz Hall, 1841 Neil Ave, Columbus, OH 43210, USA; lauren.miller257@gmail.com; 4Kansas State Veterinary Diagnostic Laboratory, College of Veterinary Medicine, Kansas State University, 1800 Denison Avenue, Manhattan, KS 66506, USA; bfeehan@ksu.edu (B.J.F.); dmarth027@vet.k-state.edu (D.M.)

**Keywords:** coronaviruses, δ-coronavirus, wild birds, epidemiology, United States

## Abstract

Porcine deltacoronavirus (δ-CoV) is the object of extensive research in several countries including the United States. In contrast, the epidemiology of δ-CoVs in wild birds in the US is largely unknown. Our aim was to comparatively assess the prevalence of δ- and γ-CoVs in wild migratory terrestrial and aquatic birds in Arkansas, Illinois, Indiana, Maryland, Mississippi, Missouri, Ohio, Tennessee and Wisconsin. A total of 1236 cloacal/fecal swabs collected during the period 2015–2018 were tested for γ- and δ-CoVs using genus-specific reverse transcription-PCR assays. A total of 61 (4.99%) samples were γ-CoV positive, with up to 29 positive samples per state. In contrast, only 14 samples were positive for δ-CoV (1.14%) with only 1–4 originating from the same state. Thus, unlike previous reports from Asia, γ-CoVs are more prevalent than δ-CoVs in the US, suggesting that δ-CoVs may spread in birds with lower efficiency. This may indicate δ-CoV emerging status and incomplete adaptation to new host species limiting its spread. Phylogenetic analysis of the partial N gene revealed that the newly identified δ-CoV strains were most closely related to the HKU20 (wigeon) strain. Further studies are necessary to investigate the role of aquatic bird δ-CoVs in the epidemiology of δ-CoVs in swine and terrestrial birds.

## 1. Introduction

Coronaviruses (CoVs) are positive-sense RNA viruses that are widespread in humans alongside various mammalian and avian species. They cause enteric, respiratory, or systemic diseases of variable severity [1,2]. CoVs belong to the *Coronaviridae* family that contains four genera: *Alphacoronavirus* (α-CoV), *Betacoronavirus* (β-CoV), *Gammacoronavirus* (γ-CoV) and *Deltacoronavirus* (δ-CoV) [3]. The virus evolves through an accumulation of point mutations and both homologous and non-homologous recombination. It is hypothesized that this ability to recombine allows the virus to evolve and create new forms which can target different species [4]. α-CoV and ß-CoV infect multiple species of mammals, but γ-CoV is only known to infect birds. In contrast, δ-CoVs are found in both birds and mammals including pigs [4]. Porcine deltacoronavirus (PDCoV) emerged in the US in 2014 as a new enteropathogenic CoV causing diarrhea, vomiting, and mortality in neonatal piglets, resulting in economic losses to the swine industry [4,5]. Its origin is unknown, but it is hypothesized that it may have spilled over from an avian host.

δ-CoV and γ-CoV are co-circulating in wild avian species, but their epidemiology (prevalence and diversity) differ [6,7,8,9,10,11,12,13]. In a screening study of 918 wild Australian birds, 141 of them tested positive for CoVs. After sequencing, δ-CoV was detected in pacific black ducks, curlew sandpiper, red-necked stints, ruddy turnstones, and pied herons [6]. However, γ-CoV (80% of sequenced samples) was identified more frequently than δ-CoV (20% of sequenced samples) [6]. γ-CoV was also detected in quails and pheasants in Italy [7]. RNA-dependent RNA polymerase (RdRp) analyses found that quail were also infected with δ-CoV [7]. Wild birds harbor genetically diverse δ-CoV with some of them potentially transmittable to pigs [8]. A previous report from Brazil showed that δ-CoVs were detected in two species of wild terrestrial birds, purple-breasted-parrot and plain parakeet. These avian CoVs were monophyletically related to CoVs from Sparrow (SPaCoV HKU19) and swine (PorCoV HKU15), which were also found to be δ-CoV. This information supports the hypothesis that δ-CoV can cross inter-species barriers, with the potential to transmit from birds to swine [8].

Previous studies of CoVs in Hong Kong reported three novel CoVs in bulbuls (BuCoV HKU11), thrushes (ThCoV HKU12) and munias (MuCoV HKU13), which were hypothesized to belong to the novel genus δ-CoV [9]. These three novel δ-CoVs clustered with a δ-CoV detected in Asian leopard cats [10]; however, further studies are necessary to understand interspecies transmission from birds [9]. In a surveillance study in Hong Kong and Cambodia, δ-CoVs were found in different wild aquatic birds: gray herons, pond herons, great cormorants, black-faced spoonbills, and several duck species [11]. δ-CoVs do not cause severe illness in birds, leading to their endemic nature in the avian population [11]. In a screening study for δ-CoVs in China, none were found in Asian leopard cats, bats, domestic cats, cattle, chickens, dogs, humans, monkeys, and rodents [12]. The investigators discovered seven novel δ-CoVs in pigs and wild birds (white-eye, sparrow, magpie robin, night heron, wigeon, and common moorhen) [12]. Genome sequencing and comparative genome analysis showed that avian and swine δ-CoVs had similar genome characteristics and structures [12]. Similarly, a recent study in the US described a novel sparrow δ-CoV that clustered together with PDCoV [13] and other terrestrial, but aquatic bird δ-CoVs.

In Brazil, screening revealed the presence of CoVs in vinaceous-breasted amazon and plain parakeet, that were closely related to δ-CoVs from birds (SpaCoV HKU17) and swine (PorCoV). This emphasizes the increased risk for direct interspecies transmission and that in contrast to waterfowl, terrestrial birds may act as intermediate hosts and are commonly found in the sites of bird-swine co-mingling [8]. In Finland, two different avian δ-CoV strains from lesser black-backed gull and black-headed gull were detected which shared 83% and 85% nucleotide identity with avian and mammalian δ-CoVs, respectively [14].

In summary, recently identified δ-CoVs are emerging globally that possess high genetic and antigenic plasticity. Porcine δ-CoV is being studied extensively in the US and elsewhere [4,5]. In contrast, the diversity and epidemiology of δ-CoVs in wild birds in the US are largely unknown. The association of porcine δ-CoV with enteric disease in pigs and its continuous spread underscore an urgent need to further investigate the mechanisms of its persistence and spread. Extensive epidemiological studies are needed to evaluate δ-CoV prevalence in avian species, swine farms, pig–bird comingling sites and high pig-traffic sites such as abattoirs, assembly yards and truck washes. In this study, we investigated the prevalence of δ- and γ-CoVs in wild migratory terrestrial and aquatic birds in Arkansas, Illinois, Indiana, Maryland, Mississippi, Missouri, Ohio, Tennessee, and Wisconsin.

## 2. Materials and Methods

### 2.1. Samples and RNA Extraction

A subset of 1236 avian cloacal/fecal swabs were selected from a collection of 16,672 that were collected for influenza A virus surveillance in different species of wild terrestrial and aquatic birds during the period 2015–2018. A total of 1236 avian cloacal swabs were collected for avian influenza virus (AIV) surveillance in different species of wild terrestrial and aquatic birds along the Mississippi Flyway, US during the period 2015–2018. Of 1236 avian cloacal swabs, 736 were from aquatic bird samples collected in Ohio, Illinois, Missouri, Maryland, Wisconsin and Tennessee in the period 2017–2018. A total of 500 were from terrestrial and aquatic bird samples collected in Ohio, Mississippi, Indiana and Arkansas in the period 2015–2016. 404 were from known species: 234 aquatic and 170 terrestrials; the remainder were from environmental fecal samples. RNA from the avian cloacal/fecal swabs was extracted using a modified commercial protocol (Ambion^®^ MagMAX™, Applied Biosystems, Foster City, CA, USA) with 50 mg/mL of bovine serum albumin (BSA) and 17% sodium sulfite.

### 2.2. γ- and δ-CoV RT-PCR Assays

A total of 1236 avian cloacal/fecal swab RNA samples were tested for γ- and δ-CoV using pancoronavirus (IN2deg/IN4deg) and deltocoronavirus-specific reverse transcription-PCR (RT-PCR) assays, respectively [15,16]. δ-CoV primers were designed using δ-CoV nucleocapsid (N) universal primers (UDCoVF: 5’-RYWGAYKSNTCNTGGTTYCA-3’ and UDCoVR: 5’-HGTGCCWGTRTARTARAAGG-3’) targeting 194bp [16]. Subsequently, after evaluating the results of Next-Generation Sequencing (NGS), we designed HKU20-specific primers targeting a 384 bp fragment of the N gene (HKU20-N-F24314 5’-TCCGCGCCTCATGGCTCTC-3’ and HKU20-N-R24698 5’-TCATGAGAAGGATTCTAG-3’). All the RT-PCR assays were performed under the same conditions using QIAgen one-step RT-PCR kit (QIAGEN Inc., Valencia, CA, USA) in a GeneAmp PCR system 9600 thermal cycler (Applied Biosystems, Foster City, CA, USA). The reaction-mixture (total 25 μL) included 5× QIAGEN OneStep buffer (5 μL), dNTP (1 μL), upstream and downstream primers (100 μmol/L, each 1 μL), RNAsin (40 U/μL, 0.25 μL), enzyme mix (1 μL), RNase-free water (10.75 μL) and RNA template (5 μL). The thermocycler protocol consisted of an initial reverse transcription step at 50 °C for 30 min, followed by PCR activation at 95 °C for 15 min, 40 cycles of amplification (95 °C for 20 s, 50 °C for 10 s, 60 °C for 30 s), and a final extension step at 72 °C for 10 min. PCR products were analyzed on 2% agarose gel. The PCR product was purified with an agarose gel QIAquick Gel extraction kit (QIAGEN Inc., Valencia, CA, USA).

### 2.3. Sequencing and Sequence Analysis

Amplicons derived using the HKU20-specific primers were sequenced at the Genomics Shared Resource of The Ohio State University (Columbus, OH, USA) by the Sanger method [17]. For NGS, previously extracted RNA underwent cDNA synthesis according to a random primer protocol performed using RevertAid H Minus First Strand cDNA Synthesis Kit (Thermo scientific, Waltham, MA, USA). PCR was conducted using True-Start DNA polymerase with 10 mM dNTPs mix and 10 pmol specific primers per reaction (Thermo Scientific), according to the manufacturer’s protocols. TruSeq Stranded Total RNA Library Prep Kit was used with 1 μg total RNA for the construction of libraries according to the manufacturer’s protocol. For rRNA-depleted library, rRNA was removed from 2.5 μg total RNA using Ribo-Zero rRNA Removal Kit (mixture 1:1 Human/Mouse/Rat probe and Bacteria probe), according to the manufacturer’s protocol (with probe concentration for epidemiology kit protocol). All cDNA libraries were sequenced using an Illumina HiSeq2000 (Illumina, San Diego, CA, USA), producing 101 × 7 × 101 bp paired end reads with multiplexing. Reads were trimmed using default parameters with CLC Genomics Workbench 8.5.1 (Qiagen Bioinformatics, Redwood City, CA, USA). Trimmed reads were de novo assembled using a word size of 64, bubble size of 100, and minimum contig length of 100. The contigs were subject to the BLASTN search. δ-CoV sequences were deposited into GenBank with the accession numbers MN379902, MN379903 and MN379904.

Sequences were assembled using BioEdit Sequence Alignment Editor and aligned using ClustalW. The N gene sequences of avian and porcine δ-CoVs obtained through BLAST search and from GenBank were included in the phylogenetic analysis. A maximum likelihood phylogenetic tree was constructed using MEGA7 software [18].

## 3. Results

### 3.1. Detection of γ-CoV in Wild Birds

A total of 61 out of 1236 birds screened were positive for γ-CoV with an apparent prevalence of 4.99% (95% confidence interval: 3.9%–6.35%). γ-CoV was identified in the states of Missouri (*n* = 29), Wisconsin (*n* = 10), Illinois (*n* = 8), Tennessee (*n* = 7), Ohio (*n* = 5), and Maryland (*n* = 2) (Figure 1A). The states where γ-CoVs were identified are shown in Figure 1A. γ-CoVs were detected in 6 different bird species: blue winged teal (*Spatula discors*) (*n* = 27), mallard (*Anas platyrhynchos*) (*n* = 16), American green-winged teal (*Anas crecca*) (*n* = 15), northern pintail (*Anas acuta*) (*n* = 1), ring-necked duck (*Aythya collaris*) (*n* = 1), and American wigeon (*Mareca americana*) (*n* = 1), all of which are aquatic bird species, meaning a prevalence of 6.3% in aquatic birds and 0% in terrestrial birds. Of interest, γ-CoVs were detected most frequently in samples from American green-winged teal, blue-winged teal and mallard that were among most frequently sampled waterfowl species (Table 1).

### 3.2. Detection of δ-CoV in Wild Birds

There were 14 of 1236 birds that screened positive for δ-CoV, corresponding with an apparent prevalence of 1.13% (95% confidence interval: 0.68%–1.91%). Thus, unlike previous reports from different countries [6,11,14], our study showed that in the US, γ-CoVs are more prevalent than δ-CoV. δ-CoVs were detected in the states of Illinois (*n* = 4), Arkansas (*n* = 4), Ohio (*n* = 2), Missouri (*n* = 2), Maryland (*n* = 1) and Mississippi (*n* = 1) (Figure 1B). The positive sampling locations are shown in Figure 2. The bird species identified were the blue winged teal (*Spatula discors*) (*n* = 6), snow goose (*Anser caerulescens*) (*n* = 4), mallard (*Anas platyrhynchos*) (*n* = 2), red-tailed hawk (*Anser caerulescens*) (*n* = 1) and northern shoveler (*Spatula clypeata*) (*n* = 1). Of these 14 positive samples, only up to 4 originated from the same state (Figure 1B), suggesting that δ-CoVs spread with low efficiency in the avian species tested. Of note, the red-tailed hawk was sampled upon intake to a wildlife rehabilitation center. Interestingly, the prevalence of δ-CoV in aquatic birds was 1.34% compared to only 0.6% in terrestrial birds, suggesting that aquatic birds represent an important natural reservoir for CoVs. Blue-winged teal and snow goose were the two species associated with the most frequent recovery of δ-CoVs (Table 1).

### 3.3. Sequence and Phylogenetic Analyses of δ-CoVs

In contrast to γ-CoVs that are widespread and have been circulating in the US for decades, the prevalence and genetic characteristics of δ-CoVs in the US are not known. To address this gap, we conducted sequence and phylogenetic analysis of the δ-CoVs identified in this study. All amplicons generated with degenerate UDCoV-specific primers were confirmed to contain δ-CoV N-gene sequences sharing ~69%–80% nucleotide identity with various avian δ-CoV species. However, due to a high number of degenerate nucleotides incorporated in these primers to allow for broad reactivity and detection of highly diverse porcine and avian δ-CoVs, the amplicons were not of satisfactory quality, with up to 10% of ambiguous nucleotides (Ns). Thus, we selected 4 δ-CoV positive samples (Table 2) that had sufficient amounts of RNA and subjected them to NGS sequencing (Table 2). For one of the samples, Blue Winged Teal coronavirus/USA/Illinois2562/2017, NGS recovered approximately 42% of eukaryotic (host genome; normal; incidental), 20% bacteria, 5% virus, and 33% other. Bacterial reads identified *Fusobacterium nucleatum* (18%), while further analyses of viral and “other” reads identified various bacteriophage (80%). Eighty-five reads were identified as δ-CoV including fragments of ORF1a/b, S, M, N and NS7a genes. Blast search and phylogenetic analysis identified that these genomic fragments shared the highest nucleotide identity (87%–93%) with Wigeon CoV HKU20, without evidence of recent recombination events (Figure 2A–C). For the other three samples, there were variable amounts of eukaryotic (2%–74%), bacterial (1%–20%), viral (5%–92%) and other/bacteriophage (5%–33%) reads; but no δ-CoV sequence was recovered. This suggests that δ-CoV RNA in these samples was present in insufficient amounts or was of low quality.

### 3.4. HKU20-Specific RT-PCR and Partial N Gene Sequence and Phylogenetic Analyses of δ-CoVs

Based on the NGS results, we designed HKU20-specific primers targeting a fragment of the N gene and re-screened the δ-CoV-positive samples for which we had sufficient amounts of RNA. Six out of 8 re-screened samples were positive in RT-PCR using HKU20-specific primers. Three samples yielded PCR fragments of sufficient quantity/quality and were used for capillary sequencing. Phylogenetic analysis of these fragments demonstrated that similar to blue-winged teal coronavirus/USA/Illinois2562/2017, the three δ-CoVs were most closely related to wigeon coronavirus HKU20 (Figure 3) sharing 83.38%–84.39% nucleotide identity in the N gene. The two samples positive in RT-PCR with UDCoV primers, but negative with HKU20 primers (Table 2) could contain insufficient amounts of δ-CoV RNA to generate a longer amplicon (targeted by the HKU20-specific primers) or could possess genetic characteristics distinct from HKU20 δ-CoV and other δ-CoVs identified in this study.

## 4. Discussion

While γ-CoVs are endemic and highly prevalent in wild and domestic birds [7,8,9,10,11,12,13], δ-CoVs have only been discovered in the US only recently. Thus, the role of wild avian species in the ecology of δ-CoVs in the US is unknown. Environmental sampling of high-risk and co-mingling sites in Alberta and Saskatchewan Canada, identified δ-CoV in sparrows, that are closer phylogenetically to porcine δ-CoVs than to those from waterfowl [19]. To date, the presence of δ-CoV is confirmed in sparrows in Canada and the US [13], in quail in Brazil [20] and in various birds in Australia [6], suggesting that δ-CoVs circulate in different avian species in the Americas. Since porcine δ-CoV often results in severe clinical disease and mortality in piglets, which impacts the swine industry, it is important to understand the morbidity and interspecies transmission rates between birds and pigs [4,5]. Due to their flocking behavior and abilities to fly long distances, birds can play a role in the dissemination of δ-CoVs among themselves and other animals [12].

The apparent prevalence of δ-CoV in our study was only 1.13%. However, this is higher compared with that observed in our previous study (0.5%) that analyzed Ohio samples from the period 2013–2014 [16]. Also, of these 14 positive samples, only up to 4 originated from the same state, suggesting that δ-CoVs spread with low efficiency in the avian species tested. Additionally, our inability to recover CoV sequences by NGS suggests that they are present in the samples at lower frequency compared with other microorganisms. The latter is more suggestive of asymptomatic carrier status as opposed to acute infection associated with clinical disease. These findings suggest that avian species likely represent a natural reservoir of δ-CoVs, while pigs and other mammals serve as spillover hosts.

The higher prevalence of γ-CoVs in the US of 4.99% is consistent with findings from some previous studies. In a screening study from Brazil, quail were susceptible to both δ- and γ-CoV [20]. The higher prevalence of γ-versus-δ-CoVs and variations in preferred avian host species have been also reported by other researchers in other countries, such as Australia [6], Asia (Hong Kong and Cambodia) [11] and Finland [14]. This may indicate δ-CoV emerging status and its ongoing spread in the US.

In this study, a higher prevalence of δ- and γ-CoVs in aquatic vs. terrestrial birds was evident, with δ- and γ-CoVs prevalence in aquatic birds being 1.34% and 6.3%, respectively, compared with only 0.6% and 0% in terrestrial birds. This suggests that aquatic birds may represent a natural reservoir for CoVs of terrestrial birds and pigs, and their concentration or survival may be increased in water sources compared with other avian habitats. Similarly, in a surveillance study in Hong Kong and Cambodia, δ-CoV was found in different wild aquatic bird species including gray herons, pond herons, great cormorants, black-faced spoonbills, and several duck species, whereas γ-CoVs were found in little whistling ducks, tufted ducks, common teals, northern shovelers, eurasian wigeons, and northern pintails [11]. In wild Australian birds, δ-CoV was detected in pacific black ducks, but it was also detected in terrestrial birds such as curlew sandpiper, red-necked stints, ruddy turnstones, and pied herons [6]. This suggests that wild birds are major reservoirs of a wide range of δ- and γ-CoVs, and the circulation of CoVs without association with clinical disease is more common than previously recognized [11]. However, it is important to note that δ-CoVs identified in terrestrial birds and pigs are more similar each other phylogenetically [12,13,21], and those from aquatic/wading birds are genetically more distinct. It is of interest, that prevalence of both γ- (92%) and δ-CoVs (86%, Table 1) was higher in colder (October–February) than in warmer (March–September) months. This indicates that, similar to previous findings on avian, animal and human CoVs, there are may be seasonal (and migration associated) fluctuations in the prevalence of avian coronaviruses in wild bird in the US [22,23,24,25].

Because phylogenetic analysis of the partial N gene of the 4 newly identified δ-CoVs (Table 2) revealed that they were most closely related to HKU20 (wigeon) δ-CoV, we hypothesized that it may be the parental strain recently introduced into the US.

In our study, these newly identified δ-CoVs were more closely related to other δ-CoV from aquatic birds, but not terrestrial avian species and pigs, which can be explained by the fact that aquatic birds occupy a separate ecological niche, while terrestrial birds and pigs may share some co-mingle sites. Although, we did not identify δ-CoV strains genetically similar to sparrow and porcine δ-CoVs, it is important to note that porcine δ-CoV outbreaks in the US were predominantly detected in the states with the highest density of pig population that in turn have a significant geographical overlap with the Mississippi Flyway (Figure 4) [26]. While a small pilot study failed to identify porcine δ-CoVs in wild waterfowl in the Mississippi Flyway [27], the overlap of the bird migratory pathways and high density swine farms creates favorable conditions for birds and pigs to exchange δ-CoV pools directly or through some other intermediate hosts.

## 5. Conclusions

In conclusion, this is the first report of the presence of HKU20-like δ-CoVs in different avian species in the US. The close relatedness of all the strains we were able to sequence to HKU20 suggests a recent introduction of δ-CoVs in the US and identification of this strain as potentially parental. We also demonstrated that aquatic birds are infected with δ-CoVs and γ-CoVs more frequently than terrestrial avian species. This, together with the observation that porcine δ-CoVs are more closely related to δ-CoVs identified in terrestrial birds, suggests that waterfowl might represent a natural reservoir for δ-CoVs. Although γ-CoVs were more prevalent than δ-CoVs, consistent with previous studies, both CoVs were detected more frequently during the cold season in our study. Further studies are necessary to investigate the role of aquatic bird δ-CoVs in the epidemiology of δ-CoVs in swine and terrestrial birds. It remains to be established: (i) if/how long avian δ-CoVs were present in South and North America prior to the porcine δ-CoV outbreaks in the period 2013–2014; (ii) which avian species represent significant natural δ-CoV reservoirs; and (iii) what is the potential of avian δ-CoVs to cross the interspecies barrier and infect swine.

## Figures and Tables

**Figure 1 viruses-11-00897-f001:**
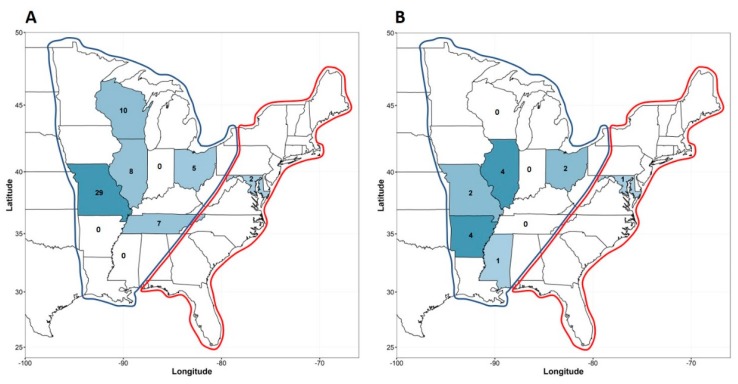
*Gammacoronavirus* (**A**) and *Deltacoronavirus* (**B**) detection in different states of the Mississippi Flyway (blue contour) and Atlantic Flyway (red contour).

**Figure 2 viruses-11-00897-f002:**
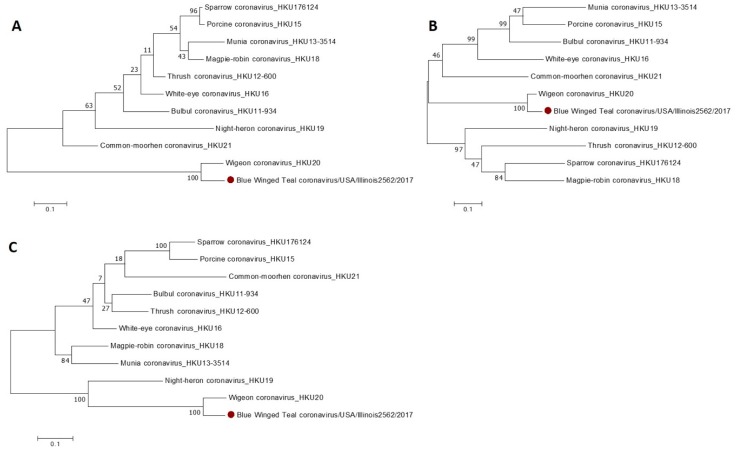
Phylogenetic trees based on partial sequences (generated using NGS) of the N (**A**), S (**B**) and M (**C**) genes of Blue Winged Teal coronavirus/USA/Illinois2562/2017.

**Figure 3 viruses-11-00897-f003:**
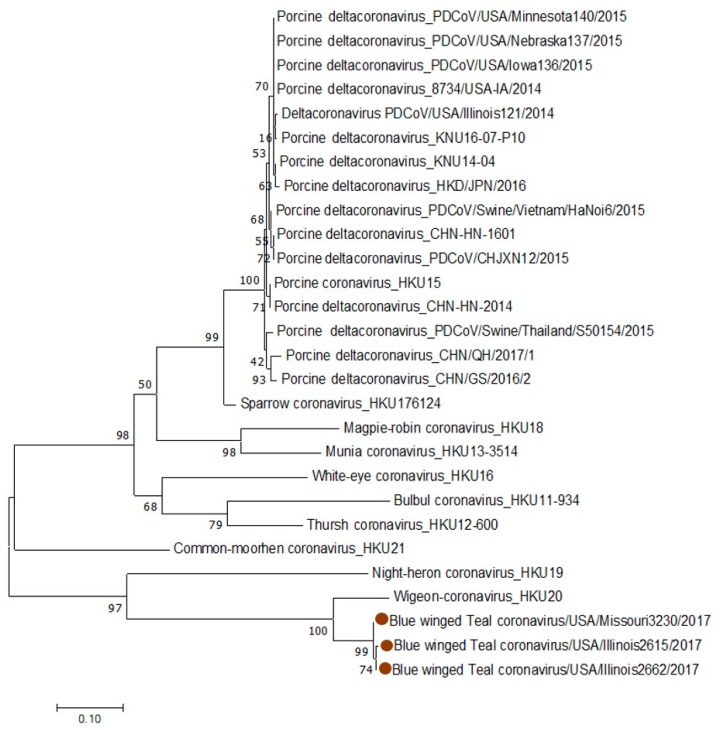
Phylogenetic trees based on partial sequences (generated using HKU20-specific primers) of the N gene of blue-winged teal coronavirus/USA/Illinois2615/2017; blue-winged teal coronavirus/USA/Illinois2662/2017 and blue-winged teal coronavirus/USA/Missouri3230/2017.

**Figure 4 viruses-11-00897-f004:**
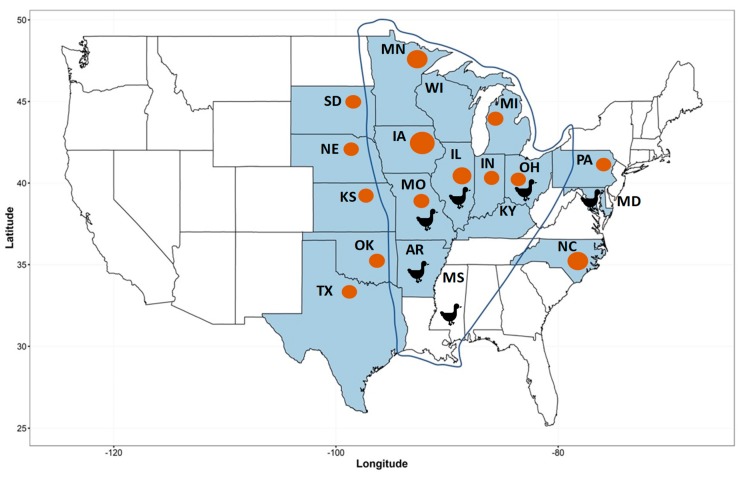
Porcine deltacoronavirus was detected in at least 18 US states (shaded in blue) along or adjacent to the Mississippi Flyway (blue contour). Orange circles mark states with the highest pig inventory [28]. Black duck icons identify the states where avian δ-CoVs were found (this study).

**Table 1 viruses-11-00897-t001:** Frequency of γ-CoV and δ-CoV in individual bird species.

Bird Species	γ-CoV+	δ-CoV+	Total
American green-winged teal	27/24.5%	0/0%	110
American wigeon	1/5.6%	0/0%	18
Blue-winged teal	27/21.4%	6/4.8%	126
Mallard	16/7%	2/0.9%	227
Northern pintail	1/8.3%	0/0%	12
Northern shoveler	0/0%	1/16.7%	6
Red-tailed hawk	0/0%	1/100%	1
Ring-necked duck	1/4.3%	0/0%	23
Snow goose	0/0%	4/5.8%	69

**Table 2 viruses-11-00897-t002:** Summary of RT-PCR and sequencing results for the 14 δ-CoV-positive samples.

Deltacoronavirus	Sample Collection Date	RT-PCR Results		Sequencing Results	

		UDCoV	HKU20	Next Generation	Partial N-Gene
Snow goose coronavirus/USA/Arkansas0009/2015	1/29/2015	+	N/A*	N/A	N/A
Snow goose coronavirus/USA/Arkansas0012/2015	1/30/2015	+	N/A	N/A	N/A
Snow goose coronavirus/USA/Arkansas0014/2015	1/30/2015	+	N/A	N/A	N/A
Snow goose coronavirus/USA/Arkansas0017/2015	1/30/2015	+	N/A	N/A	N/A
Red-tailed hawk coronavirus/USA/Ohio1248/2015	5/3/2015	+	N/A	N/A	N/A
Mallard coronavirus/USA/Ohio4381/2015	8/4/2015	+	-	Did not yield CoV sequences	N/A
Norther shoveler coronavirus/USA/Mississippi8042/2015	12/22/2015	+	N/A	N/A	N/A
Blue-winged teal coronavirus/USA/Illinois2662/2017	12/22/2015	+	+	N/A	Most closely related to HKU20
Blue-winged teal coronavirus/USA/Missouri3057/2017	10/16/2017	+	+	N/A	N/A
Blue-winged teal coronavirus/USA/Missouri3230/2017	10/23/2017	+	+	Did not yield CoV sequences	Most closely related to HKU20
Environmental coronavirus/USA/Maryland3464/2017	10/23/2017	+	-	N/A	N/A
Blue-winged teal coronavirus/USA/Illinois2537/2017	10/16/2017	+	+	N/A	N/A
Blue-winged teal coronavirus/USA/Illinois2562/2017	10/16/2017	+	+	Yielded 85 δ-CoV sequences from ORF1a/b, S, M, N/NS7a: most closely related to HKU20	Most closely related to HKU20
Blue-winged teal coronavirus/USA/Illinois2615/2017	10/16/2017	+	+	Did not yield CoV sequences	Most closely related to HKU20

* N/A—Not available: was not analyzed because there was not enough RNA.

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
