# Peer review of "Epidemiology of Deltacoronaviruses (δ-CoV) and Gammacoronaviruses (γ-CoV) in Wild Birds in the United States"

_viruses, 2019, doi:10.3390/v11100897_

Round 1
Reviewer 1 Report
The study is essentially a surveillance for the prevalence of delta and gamma coronaviruses in wildbirds (terrestrial and aquatic) from 9 states. The number of cloacal/fecal swabs screened was sufficient to represent a statistically valid sample size. The study is valid and the results support the conclusions of the paper. However, there are a few questions that the authors should answer to perhaps clarify the data and conclusions in the paper.
The Introduction reads like a literature review and could be shortened. The three (i-iii) are questions that were not really addressed in this study, so perhaps these belong in your Discussion as future areas of research? In the Abstract you state there were 61 samples positive for gamma coronavirus, with up to 29 positive samples per state. Should this be 2-9 samples per state? The primers used in the sequence and phylogenetic analysis contained degenerate nucleotides to allow for broad reactivity and detection of delta coronaviruses. Can you explain or indicate how these primers may have impacted the specificity of your reactions? Did you have a sufficient length of the N gene to sequence to have confidence in your placement of the three delta coronaviruses into a phylogenetic tree.
Overall, a reasonable study that could be strengthened by more reliable sequence results.
Author Response
Viruses-600031
We thank the editor and reviewers for insightful comments and a possibility to resubmit our manuscript. Please see below our detailed responses to each comment. All changes are highlighted in the manuscript.
RE1: The study is essentially a surveillance for the prevalence of delta and gamma coronaviruses in wildbirds (terrestrial and aquatic) from 9 states. The number of cloacal/fecal swabs screened was sufficient to represent a statistically valid sample size. The study is valid and the results support the conclusions of the paper. However, there are a few questions that the authors should answer to perhaps clarify the data and conclusions in the paper.
AU: We thank the reviewer for the overall positive comments and constructive suggestions.
RE: The Introduction reads like a literature review and could be shortened. The three (i-iii) are questions that were not really addressed in this study, so perhaps these belong in your Discussion as future areas of research?
AU: We concur with the reviewer and made attempts to shorten the Introduction and moved some text to Discussion as suggested.
RE: In the Abstract you state there were 61 samples positive for gamma coronavirus, with up to 29 positive samples per state. Should this be 2-9 samples per state?
AU: No, it is ‘up to 29 per state indeed’. If you refer to Figure 1A, you will see that 29 samples from MO were positive for gammacoronaviruses.
RE: The primers used in the sequence and phylogenetic analysis contained degenerate nucleotides to allow for broad reactivity and detection of delta coronaviruses. Can you explain or indicate how these primers may have impacted the specificity of your reactions?
AU: As mentioned, broadly reactive (degenerate) primers were necessary to identify highly heterogenous avian deltacoronaviruses (and possible carrying of porcine deltacoronaviruses). While we were able to confirm that all amplicons were deltacoronavirus-specific, when we attempted to sequence the amplicons generated using the primers, the quality of the generated sequences was low (multiple Ns) and we couldn’t determine if they were more closely related to HKU20, HKU21 or other avian deltacoronaviruses. This prompted us to attempt NGS followed by HKU20-specific (confirmatory) re-screening and sequencing of the deltacoronavirus-positive samples. The NGS and HKU20-primer based sequencing yielded highly quality results that were highly consistent between the two different methods.
RE: Did you have a sufficient length of the N gene to sequence to have confidence in your placement of the three delta coronaviruses into a phylogenetic tree.
AU: The NGS generated ~2,500bp from RdRp, 500bp from S, and ~400bp for both M and N genes, all identified as highly identical to HKU20 by Blast search and phylogenetic analysis. In agreement with the NGS results, the HKU20-specific primers yielded sequences of almost 400bp of high quality for several samples. Thus, with nearly 50% coverage of the highly conserved N gene, we are very confident about accuracy of the placement of the newly identified deltacoronaviruses in the phylogenetic tree. This placement is further supported by the Blast search.
Reviewer 2 Report
The manuscript by Paim and colleagues presents the results of screening wild bird samples for the presence of coronaviruses, followed by some characterization of the relationships of the detected viruses to previously discovered viruses. The manuscript does a very nice job describing what all of this means for the overall epidemiology of these viruses in the US, at least for the sampled regions.
Comments:
The presentation of common bird names uses a mixture of upper- and lower-case formatting (e.g. lines 51-52 versus line 57), sometimes within the same bird name (line 57). I do not know this journal’s style rule for this but suggest picking one format and using that throughout the manuscript.
On line 77, some animal species are presented as only their Latin binomial forms, which differs from previous (and subsequent).
There is a recent paper, also in Viruses (doi:10.3390/v11090768), that detected a gammacoronavirus in gulls in Canada, which might be useful to include in the Intro text section.
On line 107, suggest “environmental fecal sample”
On line 148, suggest adding the reference for MEGA
Nothing is mentioned about the samples and seasonality. Has this been considered as a variable to look at among the samples with respect to positivity?
On lines 153+ (and 166+), it would be useful to have the positive #s and the total (e.g. mallard (n=16/##))
The conclusions section does not mention anything about the gammacoronaviruses.
Author Response
Viruses-600031
We thank the editor and reviewers for insightful comments and a possibility to resubmit our manuscript. Please see below our detailed responses to each comment. All changes are highlighted in the manuscript.
RE2: The manuscript by Paim and colleagues presents the results of screening wild bird samples for the presence of coronaviruses, followed by some characterization of the relationships of the detected viruses to previously discovered viruses. The manuscript does a very nice job describing what all of this means for the overall epidemiology of these viruses in the US, at least for the sampled regions.
AU: We thank the reviewer for the positive comments, thorough revision and constructive suggestions.
RE2: The presentation of common bird names uses a mixture of upper- and lower-case formatting (e.g. lines 51-52 versus line 57), sometimes within the same bird name (line 57). I do not know this journal’s style rule for this but suggest picking one format and using that throughout the manuscript.
AU: We have checked, and the general rule appears to be not to capitalize first letters in avian species names. We have revised accordingly throughout.
RE2: On line 77, some animal species are presented as only their Latin binomial forms, which differs from previous (and subsequent).
AU: We have mostly omitted Latin binomial names throughout. Only for avian species that were positive for gamma- or deltacoronaviruses we provided both English and Latin forms.
RE2: There is a recent paper, also in Viruses (doi:10.3390/v11090768), that detected a gammacoronavirus in gulls in Canada, which might be useful to include in the Intro text section.
AU: We included the reference as suggested.
RE2: On line 107, suggest “environmental fecal sample”
AU: Re-phrased as suggested.
RE2: On line 148, suggest adding the reference for MEGA
AU: We included the reference as suggested.
RE2: Nothing is mentioned about the samples and seasonality. Has this been considered as a variable to look at among the samples with respect to positivity?
AU: We thank the reviewer for this excellent point. We now added a sentence (LL: 257-261) to address the seasonality aspect.
RE2: On lines 153+ (and 166+), it would be useful to have the positive #s and the total (e.g. mallard (n=16/##))
AU: We concur with the reviewer that this will strengthen the data presentation and included the requested information (LL150-152, 169-171 and new Table 1).
RE2: The conclusions section does not mention anything about the gammacoronaviruses.
AU: We thank the reviewer for this comment and have included statements to reflect our findings on gammacoronaviruses as well.